# Midazolam versus Dexmedetomidine in Patients at Risk of Obstructive Sleep Apnea during Urology Procedures: A Randomized Controlled Trial

**DOI:** 10.3390/jcm11195849

**Published:** 2022-10-02

**Authors:** Ivan Vuković, Božidar Duplančić, Benjamin Benzon, Zoran Đogaš, Ruben Kovač, Renata Pecotić

**Affiliations:** 1Department of Anesthesiology, University Hospital Split, 21000 Split, Croatia; 2School of Medicine, University of Split, 21000 Split, Croatia

**Keywords:** dexmedetomidine, midazolam, STOP BANG questionnaire, intraoperative complications, spinal anesthesia, sedation, obstructive sleep apnea, transurethral resection of bladder, transurethral resection of prostate

## Abstract

Benzodiazepines are the most commonly used sedatives for the reduction of patient anxiety. However, they have adverse intraoperative effects, especially in obstructive sleep apnea (OSA) patients. This study aimed to compare dexmedetomidine (DEX) and midazolam (MDZ) sedation considering intraoperative complications during transurethral resections of the bladder and prostate regarding the risk for OSA. This study was a blinded randomized clinical trial, which included 115 adult patients with a mean age of 65 undergoing urological procedures. Patients were divided into four groups regarding OSA risk (low to medium and high) and choice of either MDZ or DEX. The doses were titrated to reach a Ramsay sedation scale score of 4/5. The intraoperative complications were recorded. Incidence rates of desaturations (44% vs. 12.7%, *p* = 0.0001), snoring (76% vs. 49%, *p* = 0.0008), restlessness (26.7% vs. 1.8%, *p* = 0.0044), and coughing (42.1% vs. 14.5%, *p* = 0.0001) were higher in the MDZ group compared with DEX, independently of OSA risk. Having a high risk for OSA increased the incidence rates of desaturation (51.2% vs. 15.7%, *p* < 0.0001) and snoring (90% vs. 47.1%, *p* < 0.0001), regardless of the sedative choice. DEX produced fewer intraoperative complications over MDZ during sedation in both low to medium risk and high-risk OSA patients.

## 1. Introduction

Obstructive sleep apnea (OSA) is the most common sleep breathing disorder, characterized by repetitive episodes of complete (apnea) or partial airway obstruction (hypopnea) during sleep, often resulting in a reduction in blood oxygen saturation and usually terminated by brief arousals [1,2]. Depending on the diagnostic and sample criteria, the prevalence in the general population is estimated to vary between 3 and 24%. Still, it is generally much higher in patients undergoing surgery (24 to 41%) and in the obese and ageing populations [1,2]. OSA is considered to be a chronic disease associated with cardiovascular and metabolic consequences [3]. Therefore, patients with OSA are exposed to a higher risk for perioperative complications and represent a challenge to anesthesiologists during the entire perioperative period [4,5,6].

Today, in preoperative screening, some questionnaires are recommended, among which the STOP-BANG questionnaire is the most commonly used [4,7,8]. High-risk patients had a five times greater chance of developing unexpected perioperative complications [9].

Spinal anesthesia offers many advantages over general anesthesia, and, as such, is the technique of choice for transurethral resections of the prostate and bladder (TURB/TURP) [10], and is preferable in OSA patients [11]. In addition, sedation has increased patient satisfaction during regional anesthesia [12,13]. Still, when sedation is induced, close attention is required for potential adverse events, such as upper airway obstruction, hypoventilation, desaturation, and any cardiovascular complications [14]. Midazolam (MDZ) is one of the oldest and most familiar drugs for sedation. However, relatively stable hemodynamics can cause hypoxia, even in healthy individuals, by reducing hypoxic ventilator responses and inducing upper airway obstruction [15]. Dexmedetomidine (DEX) acts as a selective α2 adrenergic receptor agonist with sedative and analgesic effects [16]. Moreover, DEX has been less associated with the severity of OSA and respiratory depression [17]. Patients with OSA may be at increased risk for adverse respiratory events from intravenous benzodiazepine sedation. At the same time, there is a lack of evidence to assess the adverse effects of α-2 agonists in the OSA population [11].

So far, MDZ and DEX have been compared in various studies and types of surgery under regional anesthesia, showing fewer intraoperative complications in patients sedated with DEX than those with MDZ [18,19,20]. The majority of patients who underwent TURB and TURP are older and are at increased risk for OSA [7]. Therefore, the primary objective of the current study was to investigate the effects of benzodiazepine and alfa-2 agonists under spinal anesthesia on intraoperative complications such as desaturation, snoring, coughing, and restlessness in patients with TURB and TURP regarding the risk for OSA.

## 2. Materials and Methods

In this prospective, randomized, blinded clinical study, 115 adult patients aged between 18 and 80, who were scheduled to undergo elective TURB and TURP under spinal anesthesia between April 2021 and February 2022, were enrolled. The study was approved by the Ethics Committee of the University Hospital of Split (Class: 500-03/21-01/12; Registration number: 2181-147-01/06/M.S.-20-02) and was conducted under all of the ethical principles of the Seventh Revision of the Helsinki Declaration from 2013. All subjects gave their informed and individually signed consent for inclusion before participating in the study. The clinical trial registry number is NCT04817033 and can be found at Clinical-trials.gov, 20 August 2022. The exclusion criteria were as follows: regional anesthesia contraindications, American Society of Anesthesiologists (ASA) physical status classification system >III, atrioventricular cardiac block II and III degree, psychotic disorders, dementia, and participants with tracheostomy and allergy on DEX or MDZ.

The STOP-BANG questionnaire includes eight dichotomous (yes/no) questions related to the clinical features of sleep apnea (snoring, tiredness, observed apnea, high blood pressure, BMI, age, neck circumference, and male gender) [7,9]. For each question, answering “yes” scores 1, a “no” response score 0, and the total score ranges from 0 to 8. It classifies participants into three groups based on the STOP-BANG score, as follows: low (0–2), intermediate (3–4), and high risk (5–8). Those with STOP-BANG scores of 3 or 4 can be further classified as having an increased risk for moderate to severe OSA if they have both a STOP (snoring, tiredness, observed apnea, and high blood pressure) score of  ≥2 and one of the following conditions: (1) BMI  >  35 kg/m^2^; (2) neck circumference  > 40 cm; or (3) are of the male gender.

Before surgery, patients were given the STOP BANG questionnaire. Patients were stratified into two groups according to the STOP-BANG questionnaire results: high (h)-risk OSA, and low- to medium (l-m)-risk OSA. Then, patients in each OSA risk group were allocated by computer-generated permuted block randomization (block size was 10 patients) into the MDZ or DEX group. Thus, in this factorial design, four groups were created: h-risk OSA MDZ, h-risk OSA DEX, l-m risk OSA MDZ, and l-m risk OSA DEX (Figure 1). The group allocations were contained in a closed envelope that was opened before surgery after the completed enrolment procedure. 

All participants were premedicated with 5 mg of diazepam (Alkaloid, Skopje, N. Macedonia) for 12 h and 1 h before surgery. Thromboprophylaxis (enoxaparin 4000–6000 IU, Sanofi – Aventis Groupe, Paris, France) was given at least 12 h before surgery, depending on the body weight. Patients received an IV cannula with a switch for perfusor in the operating theatre. Non-invasive monitoring (electrodes for ECG, blood pressure cuff, and pulse oximeter) was placed before the induction of spinal anesthesia. The skin was disinfected, and 40 mg of 2% Lidocaine (Belupo, Koprivnica, Croatia) was given subcutaneously at the lumbar vertebrae 3/4 level. A 25 G spinal needle was used, and after the dura and arachnoid were pierced, 12.5–15 mg of 0.5% Levobupivacaine (Fresenius Kabi, Bad Homburg von der Höhe, Germany) was applied. Patients were then positioned in a uniform lithotomy position and a 9 cm high pillow was inserted. Time after a subarachnoid block was T0, and sedation with MDZ or DEX was initiated via continuous intravenous infusion. Drugs were prepared in the following manner: 50 mL of MDZ 0.3 mg/mL (midazolam, B. Braun, Melsungen, Germany) in saline for the MDZ group and 50 mL of DEX 4 μg/mL (Dexdor, Orion Corporation, Espoo, Finland) in saline for the DEX group. MDZ was initiated with a loading dose of 0.25 mg/kg/h (equivalent to 0.04 mg/kg) of ideal body mass, and DEX with loading dose of 0.5 µg/kg for 10 min. Every 10 min, the sedation level was observed with the Ramsay sedation scale (RSS) [21]. We used standard drug dosing [15] similar to other studies [18,19,20,22]. After the same loading dose, both drugs were titrated individually, that is, the reasons maintenance doses were given in intervals. Equipotency was established by titrating the drugs to achieve an RSS score of 4 or 5 (closed eyes and patient exhibited brisk or sluggish response to a light glabellar tap or loud auditory stimulus). The maintenance dose of MDZ was 0.03 to 0.06 mg/kg/h and 0.2 to 0.7 μg/kg/h for DEX. The independent blinded doctor assessed the RSS level and vital parameters every 10 min, and the primary outcomes. Patients were also blinded. Systolic, diastolic, and mean blood pressure (MAP) were noticed, along with heart rate (HR), oxygen saturation, RSS level, and adverse intraoperative events: desaturation, snoring as a sign of airway obstruction, cough, and restlessness as factors affecting the surgeon. If peripheral oxygen saturation fell below 90% for longer than 30 s, supplemental oxygen was delivered by facemask with a reservoir bag at a flow of 10 L/min. After approximately 1 min, if oxygenation was still inadequate, chin lift and jaw thrust maneuvers were performed, and an oropharyngeal airway was inserted. If the heart rate fell below 50 bpm, atropine 0.1 mg/kg (Sopharma AD, Sofia, Bulgaria) was given, and if the systolic blood pressure dropped below 100 mmHg (or MAP < 65 mmHg), ephedrine 5 mg (Sintetica, Münster, Germany) bolus was given. The total crystalloid infusion volume was noticed at the end of the surgery. All of the measurements were performed every 10 min and 1 h after surgery in the recovery room.

The sample size was estimated based on the overall rate of intraoperative complications by Silva et al. (18). α was set to 5% and power to 80%. Thus, the required sample size was 27 in both OSA risk groups. The sample size estimate was computed in G*Power software (University of Kiel, Germany).

### Statistical Analysis

The dependences of main outcome measures on the studied factors were analyzed by logistical regression. The model included OSA risk and hypnotic treatment as the only variables ($logit\left[p\left(outcome\right)\right]=\beta_{0}+\beta_{1}\cdot OSArisk+\beta_{2}\cdot drug$). Data are presented as odds ratios or proportions for categorical variables, medians, and IQRs for continuous variables. Furthermore, Mood’s test for medians and Fisher’s exact test were used to analyze the secondary outcome measures. To aid inference, 95% CI, Akaike informational criteria, and *p* values were used. *p* values were interpreted according to the ASA statement on *p* values. Graphpad Prism 9 was used as software for the statistical analysis (GraphPad Software, San Diego, CA, USA).

Primary outcome measures were as follows: (1) desaturation below 90% and (2) snoring detection regardless of duration or intensity, and (3) coughing and (4) restlessness as factors affecting surgeons because during TURB and TURP, patients have to be relaxed and calm, as their movement could result in a punctured bladder/prostate by surgical resectoscope. So, when the surgeon complained about the participant’s movement, investigators checked that on the list. Secondary outcome measures included: hemodynamic changes during sedation (heart rate and arterial blood pressure) and tobacco smoking in anamnesis.

## 3. Results

A total of 115 patients were enrolled in the present study. Four patients dropped out due to conversion of the anesthesia method from spinal to general, two of them due to patchy spinal, and two due to the obturator reflex that disrupted the surgeon (Figure 1).

The demographic and clinical characteristics in all groups were balanced and in accordance with the patient underlying conditions (Table 1). More specifically, patients with high-risk OSA were more likely to be male; high-risk OSA patients receiving DEX had a greater BMI than the same patients from the low- to medium-risk OSA group (*p* = 0.0002). In both MDZ (*p* = 0.0075) and DEX (*p* = 0.0001) subgroups, the neck circumference was also greater in high-risk OSA patients when compared with the low- to medium-risk OSA patients. DEX patients in the low- to medium-risk OSA group had smaller ASA scores than the same patients in the high-risk OSA group. Finally, in the low- to medium-risk OSA groups, patients receiving DEX needed two additional minutes to close their eyes when compared with the MDZ patients (*p* = 0.0052) (Table 1).

### 3.1. Effects of OSA Risk on Primary Outcomes

Having a high risk for OSA increased the incidence of desaturation events from 51.2% to 15.7% (OR = 8.9, 95%CI 3.25 to 28.4, *p* < 0.0001) in comparison with low- to medium-risk for OSA, regardless of sedative choice. Likewise, when snoring events were observed intraoperatively, it was found that high-risk OSA increased the frequency of snoring from 90% to 47.1% (OR = 14.26, 95% CI 4.67 to 55.58, *p* < 0.0001) when compared with low- to medium-risk OSA. Intraoperative coughing or restlessness were not influenced by OSA risk (Table 2).

### 3.2. Effects of Hypnotic on Primary Outcomes

Here, 25 out of 56 (44%) MDZ patients had desaturation events compared with 7 out of 55 (12.7%) DEX patients (OR = 0.11, 95%CI 0.28 to 0.03, *p* = 0.0001). Desaturations were resolved by only applying supplemental oxygen in the DEX group, but in MDZ group, seven patients (28% of desaturated MDZ patients) needed additional support (*p* = 0.3002). A chin lift was performed six times and the jaw thrust maneuver once in MDZ patients (three of them were high-risk OSA and four were low- to medium-risk OSA). In addition, 43 out of 56 (76%) MDZ patients snored in contrast with 27 out of 55 (49%) DEX patients (OR = 0.19, 95% CI 0.78 to 0.08, *p* = 0.0008).

DEX decreased the probability of coughing by approximately 4.5 times, as coughing was noticed in 8 out of 55 (14.5%) DEX patients vs. 24 out of 56 (42.1%) MDZ patients (OR = 0.22 95%CI 0.55 to 0.08, *p* = 0.0018). Smoking did not seem to be a risk factor for intraoperative coughing (OR = 0.82, 95%CI 0.31 to 2.11, *p* = 0.6973). The restlessness, which on its own may disrupt the surgeon, was noted in 1 out of 55 (1.8%) DEX patients as opposed to 15 out of 56 (26.7%) MDZ patients (OR = 0.049, 95%CI 0.26 to 0.002, *p* = 0.0044) (Table 2). In the MDZ group, two deliriums (2/56, 2.56%,) were observed, whereas none were observed in the DEX group.

### 3.3. Effects of Hypnotics on Hemodynamics

When it comes to hemodynamics, the maximal decreases from baseline in MAP and HR were analyzed. OSA risk did not show any significant effects on MAP or HR decrease when compared with the baseline (i.e., pre-anesthesia measurement) (Figure 2a,b). The DEX group median decrease in MAP from baseline was 26.07 mmHg (IQR 17.8 to 33 mmHg), whereas MDZ caused a median decrease of 21.45 mmHg (IQR 16.33 to 31.26 mmHg) (*p* = 0.07, Figure 2a,c). However, 15 (27%) patients in the DEX group needed ephedrine, whereas 5 (9%) MDZ patients were given the same treatment (*p* = 0.0141).

When it comes to heart rate, DEX patients experienced a median decrease of 19 bmp (IQR 13 to 23 bpm) compared with MDZ patients who had a median decrease of 12 bpm (IQR 5.2 to 18 bpm) (*p* = 0.0002, Figure 2b,d). Atropine was given to nine (16%) patients in the DEX group and only once (2%) in MDZ group (*p* = 0.0082).

## 4. Discussion

In this prospective, randomized, controlled trial, we explored if DEX or MDZ is a better sedative for TURB and TURP under spinal anesthesia regarding OSA risk. The results of our study indicate that DEX was superior to MDZ in both anesthesiologic and surgical parts, as it had fewer airway complications and patients were calmer during surgery, regardless of the OSA risk severity.

Most recent guidelines for the management of patients with OSA recommend the use of regional anesthesia when applicable [11]. During spinal anesthesia, sedation is needed as it reduces patient anxiety and may be considered as a means to increase the patient’s acceptance of regional anesthetic techniques [23]. DEX has been suggested to cause minimal respiratory depression. Many studies have shown its stable respiratory profile during spinal anesthesia [18,20,24]. On the other hand, Lodenius et al. measured upper airway collapsibility during DEX and propofol sedation in healthy volunteers and showed that DEX sedation does not inherently protect against upper airway obstruction [25]. Maybe these different effects of DEX are because of distinct loading and maintenance dose regimens. There are few studies regarding the administration of DEX in OSA patients and sedation [26,27,28,29]. Most of them compared it with propofol and showed better or similar results in respiratory effects. However, all of these sedations were in invasive procedures such as drug-induced sleep endoscopy. For sedation under spinal anesthesia, Shin et al. reported that DEX sedation was shown to be associated with a reduced incidence of obstructive airway events in patients with mild OSA compared with propofol sedation [22]. In addition, a recent study provided evidence of a positive correlation of the STOP-BANG questionnaire with oxygen saturation in patients undergoing DEX sedation, suggesting the use of the STOP-BANG score for preoperative evaluation and DEX sedation management during spinal anesthesia [30].

We found fewer desaturation episodes (i.e., airway complications) in patients treated with DEX when compared with MDZ, regardless of OSA risk, and also noticed a similar effect regarding snoring. In addition, our findings showed that high-risk OSA patients had a four times greater incidence of adverse respiratory events (23.8% vs. 5.8%) than low- to medium-risk of OSA patients treated with DEX. No eligible studies compared patients’ risk of adverse events under α2 agonists sedation with regards to confirmed OSA diagnosis [11].

During surgery, patients had to be calm and there should not have been any movement as there was a risk of perforation by the surgical resectoscope. We achieved it with moderate sedation (closed eyes, RSS 4/5), because the pain stimulus was already blocked by spinal anesthesia. It was reported that MDZ could have paradoxical reactions such as confusion, violent behavior, and restlessness [18,31,32], which is in accordance with our findings, where MDZ caused significantly more incidents of restlessness and coughing compared with DEX, which disturbed surgeons, while OSA risk had no effect on such events.

DEX acts as a selective α2 adrenergic receptor agonist and is known to cause hypotension and bradycardia due to decreased centrally mediated sympathetic tone [33]. Spinal anesthesia has a similar effect, although the lithotomy position preserves some drop in blood pressure. Although it did not reach the level of statistical significance, we did find a difference in MAP changes from baseline between DEX and MDZ groups. Furthermore, much more ephedrine was given in DEX group, and DEX showed a significantly decreased median heart rate compared with MDZ. Atropine was given more frequently to DEX patients, which corresponds with previous findings [25,34].

Similar to our results, Silva et al. showed the benefits of DEX over MDZ among older patients, but in different types of surgeries with various body positions within different regional anesthesia methods [18]. Our study was the first to investigate the advantages of different sedative choices in the context of OSA risk, in spinal anesthesia in TURB and TURP surgeries under a uniform lithotomy position (with a 9 cm high pillow) with both sedatives administered via continuous intravenous infusion without the use of opioids. This study proposes the use of the DEX over MDZ in sedation management for TURB and TURP surgeries.

This study has some limitations. It is well known that polysomnography remains the gold standard in sleep investigation and the diagnosis of OSA. Because of the limitations for performing polysomnography in our Split Sleep Center during the COVID pandemic, we could evaluate the risk of OSA for patients using the STOP-BANG questionnaire. Snoring was observed as an indication of upper airway obstruction in our study. However, it is very difficult to precisely define the presence of snoring due to the different intensities, durations, and level of obstructions, which indicates the need for standardization in clinical trials. In addition, one of limitations is that this study was done in one center and the results could be different in centers not premedicating before such procedures. We only followed up patients during the intraoperative period, hence it would be interesting to prolong that follow up in future studies, so that outcomes such as risk of dysuria due to atropine administration, delirium, and time spent in hospital can be observed.

In conclusion, DEX sedation was shown to be associated with a reduced incidence of airway complications and patients were calmer with less surgery-disturbing factors of restlessness and coughing in comparison with MDZ sedation, for both the low- to medium-risk and high-risk OSA patients. Although DEX-treated patients showed more hemodynamic instability, this was easily resolved by medications and thus we recommend DEX as the sedative of choice for TURB and TURP under spinal anesthesia.

## Figures and Tables

**Figure 1 jcm-11-05849-f001:**
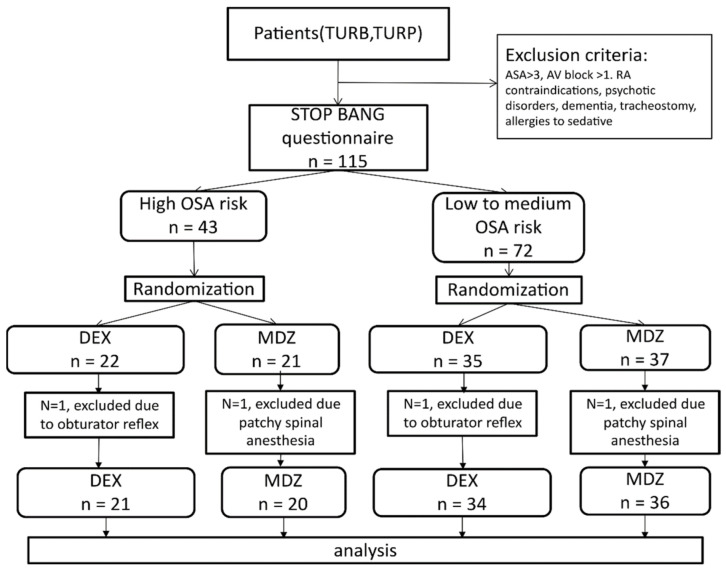
Patient flowchart. OSA—obstructive sleep apnea; TURB—transurethral resection of bladder; TURP—transurethral resection of prostate; RA—regional anesthesia; ASA—American Society of Anesthesiologist; DEX—dexmedetomidine; MDZ—midazolam.

**Figure 2 jcm-11-05849-f002:**
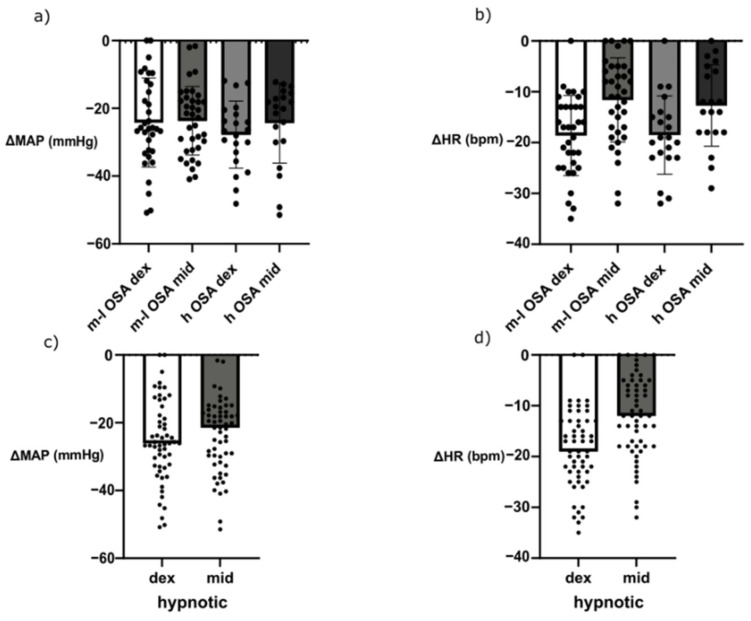
Hemodynamic outcomes. Maximal decrease from the baseline (i.e., pre-anesthesia measurement) of the mean arterial pressure (MAP) and heart rate (HR) in patients stratified by OSA risk and hypnotic; mean and SD are also presented (**a**,**b**). Maximal decrease from baseline of MAP and HR in patients stratified by hypnotic only; median is also presented (**c**,**d**).

**Table 1 jcm-11-05849-t001:** Demographic and clinical data.

OSA Risk	Low to Medium				High				*p* Value *
hypnotic	DEX		MDZ		DEX		MDZ		
age (years, median, IQR)	65	61 to 68.25	69.5	61.25 to 69.5	68	63 to 68	69	67 to 71.75	0.1207
sex (male, %)	67.65		80.56		90.48		100		0.0169
BMI (kg/m^2^, median, IQR)	25.75 ^a^	23.4 to 29.56	28.5	24.59 to 30.03	30.79 ^a^	27 to 33.93	28.72	27.44 to 33.29	0.0009
neck circumference (cm, median, IQR)	39 ^b^	36.5 to 44	41 ^c^	37.5 to 45	46 ^b^	42.5 to 48	45 ^c^	42.25 to 47	<0.0001
ASA score (median, IQR)	2 ^d^	2 to 2	2	2 to 2	2 ^d^	2 to 3	2	2 to 2	0.0097
duration of surgical procedure (min., median, IQR)	50	30 to 70	50	30 to 70	40	30 to 70	60	40 to 70	0.3332
time for eyes closing (min., median, IQR)	10 ^e^	9 to 11.5	8 ^e^	7 to 9	9	7 to 11	9.5	8 to 10.75	0.0389
baseline SpO_2_ (%, median, IQR)	97 ^f^	96.75 to 99	98 ^g^	97 to 98	96 ^f,g^	95 to 97	97	96 to 99	0.0053
STOP BANG (point, median, IQR)	3 ^h^	2 to 3	3 ^i^	2 to 4	5 ^h,i^	5 to 6	5 ^h,i^	5 to 6	<0.0001

* Kruskal–Wallis test for continuous data and chi square test for categorical data, omnibus *p* value. DEX—dexmedetomidine; MDZ—midazolam; BMI—body mass index; ASA—American Society of Anesthesiologists; OSA—obstructive sleep apnea; ^a,b,c,d,e,f,g,h,i^—Dunn’s post hoc test, *p* < 0.05.

**Table 2 jcm-11-05849-t002:** Intraoperative complications.

Outcome	OSA Low to Medium Risk				OSA High Risk				OR (OSA High)	95% CI		OR (Dex)	95% CI	
	DEX (Total *n* = 34)		MDZ (Total *n* = 36)		DEX (Total *n* = 21)		MDZ (Total *n* = 20)			Lower	Upper		Upper	Lower
	*n*	%	*n*	%	*n*	%	*n*	%						
desaturation	2	5.88%	9	25.00%	5	23.81%	16	80.00%	8.981	3.257	28.4	0.112	0.032	0.323
snoring	10	29.41%	23	63.89%	17	80.95%	20	100.00%	14.26	4.67	55.58	0.195	0.072	0.491
coughing	4	11.76%	14	38.89%	4	19.05%	10	50.00%	1.62	0.66	3.98	0.225	0.084	0.548
restlessness	0	0.00%	9	25.00%	1	4.76%	6	30.00%	1.56	0.48	5.01	0.049	0.002	0.261

OSA—obstructive sleep apnea, DEX—dexmedetomidine, MDZ—midazolam, OR—odds ratio.

## Data Availability

The datasets are available upon reasonable request to the corresponding author.

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
