# Peer review of "Midazolam versus Dexmedetomidine in Patients at Risk of Obstructive Sleep Apnea during Urology Procedures: A Randomized Controlled Trial"

_jcm, 2022, doi:10.3390/jcm11195849_

Round 1

Reviewer 1 Report

Comment

1. (Page 1, line 23-25) The sentences of “Having high risk for OSA increased the incidence rates of desaturation (15.7% vs 51.2%, p<0.0001) and snoring (47.1% vs 90%, p<0.0001) regardless of the sedative choice” is incorrect – it is inverted. Should be (51.2% vs 15.7%).

2. Why are there more spaces in the manuscript?

3. The manuscript of “Introduction” is about too long; it could be shortened substantially.

4. Please provide randomization procedures information for readers. Was stratified randomization used?

5. The authors stated that (Page 3, line 122-125) “If peripheral oxygen saturation fell below 90% for longer than 30 seconds, supplemental oxygen was delivered by facemask with reservoir bag at the flow of 10 L/min. If oxygenation was still inadequate”, how long it would be observed for the procedures of supplemental oxygen? 

In addition, peripheral oxygen saturation fell below 90% for longer than 30 seconds, is there a minimum oxygen saturation limit here? such as peripheral oxygen saturation fell below 90% for longer than 30 seconds or fell below 85%? There is a high probability of a sharp drop in patient oxygen saturation over a 30-second period.

6. “All measurements were performed every 10 minutes and 1 hour after surgery in urology intensive care.” Why is urology intensive care? it's confusing.

7. Authors should report the results of a randomized controlled trial must follow the CONSORT statement. The study needs to re-ensure that adequate information is provided, including randomization, blinding, and definition of outcome measures.

8. The sample size calculation part should consider the correction of multi-group comparison, why still choose 0.05? (Page 3, line 130-133).

9. The study is underpowered for the analyses, and no adjustment was made from multiple comparisons; this should be approached with extreme caution.

10. (Page 3-4, line 142-149). Clarify – was each adverse event analyzed separately or as a composite?

11. (Page 3-4, line 159-170). The authors stated that Demographic and clinical characteristics in all groups were balanced (Table 1). However, significant difference was found in BMI, neck circumference, ASA score, et. al. Such a description is strange and contradictory. why is this? Scientific papers should accurately describe the results.

The production of Table 1 also lacks anti-readability and is not friendly to readers.

11. (Page 5, line 172-173). Throughout the paper --- it is inverted. Should be (51.2% vs 15.7%).

12. Important considerations:

The authors achieved imprecision resulted in a width CI, ranging from 4.67 to 55.58 or 3.25 to 28.4. Again, this emphasized that the study is underpowered for the analyses. The confidence interval is too wide, the results should be interpreted with caution.

13. I have questions about the authenticity and accuracy of the results (line 181-196).

Say “(OR=0.11, 95%CI 0.28 to 0.03); (OR=0.19, 95% CI 0.78 to 0.08, p=0.0008); (OR=0.22 95%CI 0.55 to 0.08, p=0.0018); (OR= 0.049, 95%CI 0.26 to 0.002, p=0.0044)”

How can this happen? The upper limit of the confidence interval is less than the lower limit? There are major problems in data analysis and data interpretation in the manuscript.

14. Are all patients receiving oxygen routinely? Or only in special cases oxygen? What is the patient's basal oxygen saturation?

15. How to deal with baseline imbalance? Subgroup analysis?

16. References: Many studies have found a high percentage of errors in reference lists. It should follow the instructions for authors and maintain uniformity (Issues with capitalization, punctuation, page numbers, etc.).

Reviewer 2 Report

Vuković et al. conducted a RCT to compare dexmedetomidine (DEX) and midazolam (MDZ) sedation on intraoperative complications during transurethral resections of the bladder and prostate regarding the risk for OSA. I have several concerns.

1. A diagnosis of OSA is associated with a higher risk of perioperative complications (PMID: 31986433; PMID: 35332537). Recent meta-analysis (PMID: 35217467) also demonstrated that a high risk of OSA, as assessed using the STOP-Bang questionnaire, was associated with a higher incidence of postoperative complications.

2. The manuscript need to discuss the differences of the present study with previous ones, such as (PMID: 30777901).

3. Could the author clarify who were blinded? Did the anesthetists who adjusted the infusion rate of DEX and MDZ were blinded?

4. Could the author provide the results of statistical analysis between l-M OSA+DEX with l-M OSA+MDZ, H OSA+DEX with H OSA+MDZ in Table 1, so we can know whether the baseline was balanced.

5. For elderly male patients, use of atropine may increase the risk of dysuria. Since DEX group has more patients using atropine and may lead to more risk of dysuria, this need to be discussed.

Author Response

Response to Review 2

Thank you for your detailed review of our manuscript and your valuable comments. In the lines below we have tried to address all of the issues pointed out.

Vuković et al. conducted a RCT to compare dexmedetomidine (DEX) and midazolam (MDZ) sedation on intraoperative complications during transurethral resections of the bladder and prostate regarding the risk for OSA. I have several concerns.

  1. A diagnosis of OSA is associated with a higher risk of perioperative complications (PMID: 31986433; PMID: 35332537). Recent meta-analysis (PMID: 35217467) also demonstrated that a high risk of OSA, as assessed using the STOP-Bang questionnaire, was associated with a higher incidence of postoperative complications.

Thank you for suggestion. We inserted both articles according to your suggestion in our manuscript.

  1. The manuscript need to discuss the differences of the present study with previous ones, such as (PMID: 30777901).

Thank you very much for your suggestion. We compared our study with (PMID: 30777901) in discussion section (lines 259 – 266).

  1. Could the author clarify who were blinded? Did the anesthetists who adjusted the infusion rate of DEX and MDZ were blinded?

Patient and a physician who assessed depth of sedation (Ramsay score of 4/5) and outcomes were blinded. Anaesthetist, who adjusted the infusion rate of DEX and MDZ, was not blinded due to legal and ethical reasons and was not part of the study. (line 112 -. 113)

  1. Could the author provide the results of statistical analysis between l-M OSA+DEX with l-M OSA+MDZ, H OSA+DEX with H OSA+MDZ in Table 1, so we can know whether the baseline was balanced.

These comparisons were done by Dunns post hoc test and if there are differences with p<0,05 they were marked with superscript. For your particular comparison the only difference was in eye closing in low to medium OSA risk group which we explained in line 270 – 273.

  1. For elderly male patients, use of atropine may increase the risk of dysuria. Since DEX group has more patients using atropine and may lead to more risk of dysuria, this need to be discussed

Thank you for your observation. We addressed this matter in discussion section (lines 276 - 279). Unfortunately we only included intraoperative period in which dysuria couldn't be observed because patients are under spinal anesthesia, so future studies could investigate this interesting question.

Reviewer 3 Report

A nice RCT on a subject applicable in a lot of hospitals.

The premedication needs further discussion, did this affect the results. 

How long was the follow up, according to delirium? Any difference in "hangover"? Time spent in hospital?

Author Response

Response to Reviewer 3:

Thank you for your detailed review of our manuscript and your valuable comments. In the lines below we have tried to address all of the issues pointed out.

A nice RCT on a subject applicable in a lot of hospitals.

Thank you for this nice comment.

The premedication needs further discussion, did this affect the results.

We only gave 5mg oral Diazepam one hour before surgery to both DEX and MDZ groups. It is low anxiety dose that can't cause respiratory depression and recommended premedication dose is 10mg intramusculary (PMID: 30725707), so such a low dose shouldn't affect the results. We adressed this issuein the discussion section lines 218 – 221.

How long was the follow up, according to delirium? Any difference in "hangover"?

Time spent in hospital?

Thank you for your observation. We addressed this matter in discussion section (lines 276 - 279). Unfortunately, in our study we only included intraoperative period so future studies should investigate this interesting question.

Round 2

Reviewer 1 Report

The author does not address my addition to the lack of methodological detail, and I remain skeptical of statistical validity.

Author Response

Response to the Reviewer 1

The author does not address my addition to the lack of methodological detail, and I remain skeptical of statistical validity.

Thank you for your opinion. After your first review, we clarified in the Methods sections the study design. Furthermore, we uploaded the CONSORT checklist with the manuscript, and Figure 1 is a CONSORT flowchart. We are very sorry that you remain skeptical about statistics. Both study design and data analysis were done according to the principles in Altman’s textbook (1). Finally, let us leave it to the readership to assess trustworthy of our findings; in this vein, we made a fully transparent presentation of data and study design in the manuscript.

  1. Altman D. R. Chapters 5 and 12.5 In: Practical statistics for Medical Researcher, Chapman & Hall, 2018
